# Latent Shattering: Turning Unconditional Pretrained Generators Into Conditional Models By Imposing Latent Structure

## Abstract

Deep generative models, such as GANs and VAEs, have gained substantial attention for their ability to synthesize realistic data. Pretrained generative models are often *unconditional*, thus do not easily allow the user to specify the *class* of the output. Yet supporting conditional generation offers inherent benefits for many tasks. Due to current models requiring huge data sets and often prohibitively expensive computational resources for training, it is desirable to have a lightweight method that can convert pretrained unconditional generators into conditional models without retraining. Previous research into this problem is limited, typically assuming either access to classifiers that identify which regions of the generator's latent space correspond to specific classes, access to labeled data, or even retraining of the generative model itself. These strict requirements pose a serious limitation. In this work, we propose LASH, a fresh approach at the conversion of unconditional generators into conditional models in a *completely* unsupervised manner *without* requiring retraining nor access to *any* real data. Instead, the key principle of LASH is to identify points in the generator's latent space that are mapped to low-density regions of the output space. The insight is that by removing these points, LASH "shatters" the latent space into distinct clusters where each cluster corresponds to a semantically meaningful mode in the output space. We demonstrate that these modes correspond to distinct real-world classes. Lastly, LASH utilizes a simple Gaussian mixture model to adaptively sample from these clusters, supporting unsupervised conditional generation. Through a series of experiments on MNIST, FashionMNIST, and CelebA, we demonstrate that LASH significantly outperforms existing methods in unsupervised conditional sampling.

## 1 Introduction

**Motivation.** Generative models are increasingly being adopted in a wide range of domains due to their impressive ability to synthesize realistic data for many modalities (Bond-Taylor et al., 2022). Many popular classes of generative models, such as Generative Adversarial Networks (GANs) (Goodfellow et al., 2014) and Variational Auto-Encoders (VAEs) (Kingma & Welling, 2013), utilize a *generator*[1] that maps from a *lower*-dimensional *latent space* $\mathbb{Z}$ to a *higher*-dimensional *data space* $\mathbb{X}$. Though some pretrained GAN and VAE models allow for conditional generation, many are *unconditional* (Laria et al., 2022); meaning the user does not have control over which *class* is sampled from. For instance, a user can use an unconditional generator trained on a dataset of pictures on animals to synthesize random images of animals, but cannot specify that the image be a picture of a "dog". Due to the expense and difficulty associated with training these generative models (Chen, 2021), it would be beneficial to have the ability to convert pretrained unconditional models into conditional models *without* having to retrain the generator or any other deep network.

**State-of-the-Art.** Existing work in performing class-specific generation given pretrained unconditional models typically have stringent restrictions. Namely, they either require 1) classifiers or *energy functions* that provide feedback on which class each point in the latent space $\mathbb{Z}$ corresponds

---

[1] In auto-encoder based models, the generator corresponds to the decoder network

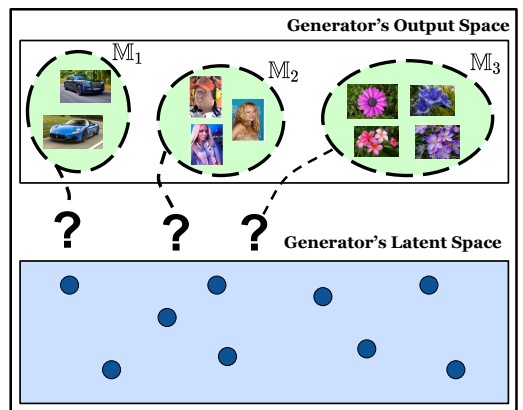 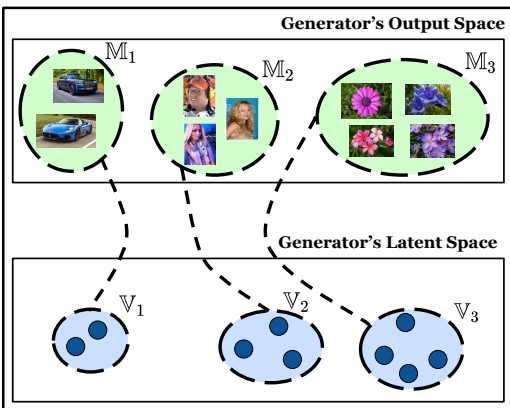

(a) An example of an unstructured latent space.      (b) An example of a structured latent space.

Figure 1: a) The latent space of the generator typically follows a data-independent prior and is thus *unstructured*; clusters in the output space do not have corresponding clusters in the latent space. b) Alternatively, a structured latent space consists of distinct clusters that each map to a unique sub-manifold in the data space. Our LASH approach produces a structured space.

to when passed through the generator (Wu et al., 2022; Engel et al., 2018); or 2) labeled data and training of additional deep networks to guide the generator's output (Laria et al., 2022). However, neither access to additional networks that can provide this class-specific feedback nor access to training data and sufficient compute resources is practical in many real-world resource-constrained settings.

**Problem Statement.** This leads to the open research question tackled by this work, namely, *can we convert a pretrained generator into a conditional model in an unsupervised manner and without retraining any deep networks?* To the best of our knowledge, we are the first to study this open problem. For this work, we limit our focus to generative models that map from a lower dimensional latent space to a higher dimensional data space, as is typical for models such as GANs and VAEs (Goodfellow et al., 2014; Kingma & Welling, 2013).

**Challenges.** Performing class-conditional generation is extremely challenging, especially when we assume we have no labels nor auxiliary classifiers at our disposal to provide supervision. How can we possibly determine what classes even exist in the generator's output space? In this work, we put forth that a natural proxy for class labels is to identify the different modes or sub-manifolds in the data space. This is motivated by recent work which has provided evidence that high-dimensional data such as images often lay on a union of disjoint sub-manifolds (Brown et al., 2023), where these sub-manifolds correspond to semantically meaningful classes. For example, in MNIST each digit lies on a different sub-manifold in the image space (Brown et al., 2023). Therefore, a potential solution may aim to identify which regions of the latent space $\mathbb{Z}$ correspond to each of these sub-manifolds, and then selectively sample from these regions. However, this is difficult because the distribution over the latent space of leading generative models is typically enforced as a prior and is thus *not data-dependent* (Goodfellow et al., 2014; Kingma & Welling, 2013); e.g., the distribution is Gaussian or uniform. In other words, there is no fluctuation in density or any other variation that would align with any of the submanifolds in the data space.

**Our Proposed Approach: LASH.** In this work, we propose **La**tent **Sh**attering (LASH), the first approach to convert pretrained unconditional generators into conditional models. The key idea of LASH hinges on the fact that common generative models utilize a continuous function with a connected latent space, and are therefore unable to produce perfectly distinct sub-manifolds in their output space (Tanielian et al., 2020). Assuming the true data generation is concentrated on a set of sub-manifolds, a well-trained generator will place most mass on these sub-manifolds but provably *must*[2] also place a small amount of mass *between* the manifolds (Khayatkhoei et al., 2018). One

---

[2]Assuming the generator is a continuous function, which nearly alway holds.

of our key insights is thus to use knowledge of the density of the latent space's prior along with a calculated volume change induced by the generator to identify regions of the latent space that are mapped to these low-density off-manifold regions in the data space. By removing the latent points that exist in these low-density regions, we can *shatter* the latent space into $k$ sub-manifolds, each of which corresponds to one of the $k$ manifolds in the data space. This would then allow us to learn to sample from and compute the density for each of these $k$ manifolds by fitting a parametric distribution to each manifold. An illustration highlighting the difference between the standard generator latent space and the structured space produced by LASH is depicted in Figure 1.

**Contributions**   In this work, we:

- Propose the Latent Shattering (LASH) approach to split the latent space of pretrained generative models into meaningful disjoint manifolds, by identifying and removing instances mapped between modes in the output space.

- Utilize LASH to learn to sample from each of these disjoint manifolds individually, providing the first solution for turning a pretrained unconditional generative model into a conditional model **in a completely unsupervised manner** - with no external guidance.

- Perform a series of experimental studies that demonstrate that our proposed approach routinely identifies semantically meaningful regions of the generator's latent space *without* requiring any additional data, labeled or otherwise.

## 2   PROBLEM STATEMENT

Let $G : \mathbb{Z} \to \mathbb{R}^N$, $\mathbb{Z} \subset \mathbb{R}^M$ with $M < N$, be a pretrained generative model such that $G$ is a smooth continuous function. Let $\mathbb{Z}$ be a simply connected manifold. Let $G(\mathbb{Z})$ be a Riemannian manifold with intrinsic dimensionality $M$, and let $q_G$ be the "generated distribution" defined as a *pushforward distribution* $q_G = G_\# p_{\mathbf{z}}$, where $p_{\mathbf{z}}$ is the prior "noise" distribution over $\mathbb{Z}$. $p_{\mathbf{z}}$ here is a Gaussian or uniform distribution (Bond-Taylor et al., 2022), but we do not require this to be the case.

Further, assume that $G$ was trained to minimize a divergence between $q_G = G_\# p_{\mathbf{z}}$ and $p_{\mathbf{x}}$, where $p_{\mathbf{x}}$ is the target distribution of real data. Let $\mathbb{X} \subset \mathbb{R}^N$ be a subset of $\mathbb{R}^N$ where the real data lies. Let $\mathbb{X}$ consist of $k$ distinct non-overlapping sub-manifolds: $\bigsqcup_{i=1}^{k} \mathbb{M}_i = \mathbb{X}$, and $(\mathbb{M}_i \cap \mathbb{M}_j = \emptyset) \, (\forall i \neq j)$.

Our goal is to find a latent space $\mathbb{Z}^* \subset \mathbb{Z}$ and a corresponding distribution $q_{\mathbf{v}}$ with the following properties: 1) $q_{\mathbf{v}}$ has minimal divergence from $q_{\mathbf{z}}$; 2) the latent space $\mathbb{Z}^* = \bigsqcup_{i=1}^{k} \mathbb{V}_i$, $\mathbb{V}_i \cap \mathbb{V}_j = \emptyset$ $\forall i \neq j$ consists of $k$ distinct non-overlapping manifolds $\mathbb{V}_i$; and 3) if latent code $v_i \in \mathbb{V}_i$ then $G(v_i) \in \mathbb{M}_i$. See Figure 1 for an illustration of this goal. In effect, if there are $k$ disjoint sub-manifolds in the data distribution, then we want to find a latent distribution that likewise has $k$ disjoint sub-manifolds - with a one-to-one correspondence between them.

To develop our method, we make use of a small set of assumptions that we state below.

**Assumption 1.** *$G$ is an injective, continuous, differentiable mapping from $\mathbb{Z}$ to $\mathbb{R}^N$. Thus, the pullback distribution $(G^* q_G)(G(\mathbf{z}))$ is well-defined for all $\mathbf{z} \in \mathbb{Z}$ and $G(\mathbf{z})$ is unique for all $\mathbf{z} \in \mathbb{Z}$, and $\{G(\mathbf{z}) \mid \mathbf{z} \in \mathbb{Z}\}$*

Assumption 1 requires $G$ to be a one-to-one mapping from the latent space to the data space, such that each generated instance has a unique latent code. This assumption is commonly used by other works (Abbasnejad et al., 2019; Humayun et al., 2022b). Like them, we do not practically require that this exactly holds for every instance and we do not need to explicitly define a left-inverse; injectivity is required for theoretical analysis, and in practice our approach will be effective so long as it holds for almost all of the input space.

**Assumption 2.** *There exists a positive real valued number $\rho$ such that $p_G(G(\mathbf{z})) > \rho$ if $G(\mathbf{z}) \in supp(p_{\mathbf{x}})$, and $p_G(G(\mathbf{z})) \leq \rho$ if $G(\mathbf{z}) \notin supp(p_{\mathbf{x}})$ otherwise.*

In essence, Assumption 2 states that the density of the generated distribution is higher when on-manifold with regards to the real data $\mathbb{X}$, and lower when off-manifold. For example, realistic synthetic data points have higher density than unrealistic synthetic points.

**Assumption 3.** *For each submanifold in the data space, $\mathbb{M}_i \in \mathbb{X}$, $\exists \, \mathbf{z} \in \mathbb{Z}$ such that $G(\mathbf{z}) \in \mathbb{M}_i$.*

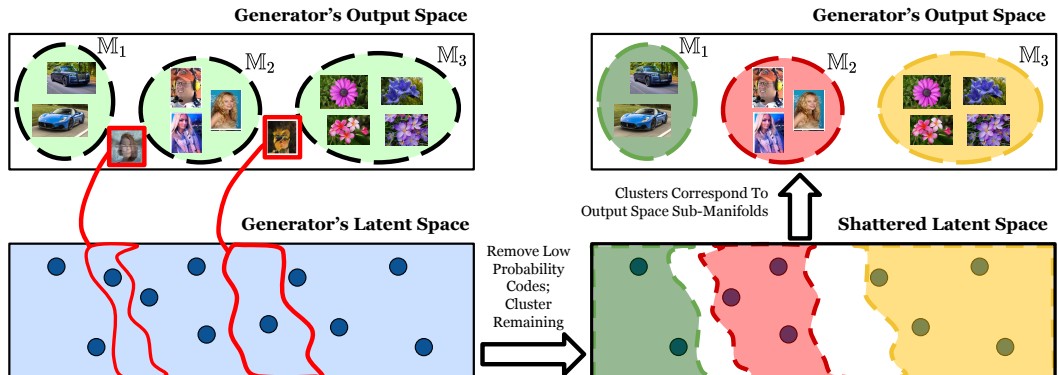

Figure 2: *Latent Shattering*. We remove latent codes that correspond to low-probability (*off-manifold*) samples in the output space, and then cluster the higher-probability latent codes. This results in a disjoint, structured latent space.

Assumption 3 states that the generator places at least some mass on each mode or submanifold of $\mathbb{X}$. For example, if the real data consists of 10 distinct classes, the generator $G$ would be capable of producing some instances of each class.

## 3 METHODOLOGY

In this section, we describe our approach to transform the "unstructured" single-manifold latent space into $k$ distinct sub-manifolds, where each of the k submanifolds contains latent codes that are mapped to only one single submanifold in the target data space $\mathbb{X}$. Section 3.1 describes this approach of finding these $k$ submanifolds in $\mathbb{Z}$. Thereafter, we describe our proposed approach for selectively sampling from each of these submanifolds in Section 3.2.

### 3.1 LATENT SHATTERING: DIVIDING LATENT SPACE INTO MEANINGFUL SUBMANIFOLDS

Note that as $G$ is a continuous function and the latent space $\mathbb{Z}$ is simply connected, then $\mathbb{G} = G(\mathbb{Z})$ will likewise be simple connected (Tanielian et al., 2020). Thus, the generated data will by default lie on a single generated manifold in $\mathbb{R}^N$. However, as $G$ was trained to match the distribution of real data (which lies on $k$ submanifolds), $q_G$ should place more mass on the submanifolds in $\mathbb{X}$ and lower mass between submanifolds (see Assumption 2). This leads us to define the property of a generated manifold being $k$-$\rho$-*disconnected*:

**Definition 1** ($k$-$\rho$-disconnected)**.** *For a generative model $G : \mathbb{Z} \to \mathbb{R}^n$, let $\mathbb{G}$ be the support of the generated distribution $q_G$. Let $\mathbb{G}_{\backslash\rho} \subset \mathbb{G}$ be a subset of the support such that $\mathbb{G}_{\backslash\rho} = \{\boldsymbol{x} \in \mathbb{G} | q_G(\boldsymbol{x}) \geq \rho\}$. Then, we say that $G$ is $k$-$\rho$-**disconnected** if $\mathbb{G}_{\backslash\rho}$ consists of $k$ disjoint submanifolds; i.e., $\mathbb{G}_{\backslash\rho} = \bigsqcup_{i=1}^{k} \mathbb{S}_i$ and $\mathbb{S}_i \cap \mathbb{S}_j = \emptyset \, \forall \, i \neq j$.*

Definition 1 means that we say the manifold of generated data is $k$-$\rho$-disconnected if removing points where $q_G$ has a density less than $\rho$ results in a set $\mathbb{G}_{\backslash\rho}$ that consists of $k$ disjoint submanifolds.

Next, we define the density of the generated data after removing the regions with density $< \rho$ as the $\rho$-shattered density $q_{G\backslash\rho}$.

**Definition 2** ($\rho$-shattered density $q_{G\backslash\rho}$)**.** *For a generative model $G : \mathbb{Z} \to \mathbb{R}^n$, let $\mathbb{G}_{\backslash\rho} = \{\boldsymbol{x} \in \mathbb{G} | q_G(\boldsymbol{x}) \geq \rho\}$. The density of the $\rho$-**shattered density** $q_{G\backslash\rho}$ is the density function defined over $\mathbb{G}_{\backslash\rho}$ such that $q_{G\backslash\rho}(\boldsymbol{x}) = \left(1 - \int_{\mathbb{G}-\mathbb{G}_{\backslash\rho}} q_G(\boldsymbol{x}')d\boldsymbol{x}'\right)^{-1} q_G(\boldsymbol{x})$.*

Definition 2 simply defines a distribution over the disjoint generated space $\mathbb{G}_{\backslash\rho}$ as equivalent to the original generated density $q_G$ with only a few slight modifications: 1) the density is set to 0 where $q_G < \rho$; and 2) the remaining non-zero density is scaled by a constant factor to account for the mass that was "removed".

We can begin to see where these two definitions are useful when we consider Lemma 1.

**Lemma 1.** *If $\mathbb{X}$ consist of $k$ disjoint submanifolds $\mathbb{M}_1$ to $\mathbb{M}_k$, then $\exists \rho$ such that $G$ is $k$-$\rho$-disconnected.*

The above lemma follows from Assumptions 1 - 3. This means that if $G$ was trained to match a real-data distribution over $k$ submanifolds, then there should exist a density value for the generated distribution that distinguishes on-manifold instances from off-manifold instances. Put differently, there is a value $\rho$ such that the generated density is greater than $\rho$ for all generated data that lies on any of the real-data manifolds and is less than $\rho$ otherwise.

In effect, the preceding statements have lead up to the insight that we can remove low-probability generated instances (where low-probability is defined in reference to a scalar value $\rho$) such that the remaining generated data will lie on $k$ distinct submanifolds. However, this does not tell us anything about the struture of our latent space - which was our initial goal. To that end, we now introduce the following important theorem underlying our proposed method.

**Theorem 1.** *Let $G$ be $k$-$\rho$-disconnected by Definition 1. Then the support of the pullback distribution $(G^* q_{G \backslash \rho})(G(\boldsymbol{z}))$, $\mathbb{Z}_{\backslash \rho} = \{\boldsymbol{z} \in \mathbb{Z} \mid q_{G \backslash \rho}(G(\boldsymbol{z})) > 0\}$ consists of $k$ disjoint submanifolds.*

Theorem 1 states that if removing generated instances with a density less than $\rho$ not only makes the generated distribution lie on $k$ *disjoint submanifolds in the data space*, but also results in a *latent space with $k$ distinct submanifolds*.

The following lemma and theorem now establish the *relationship* between the submanifolds in $\mathbb{Z}$ to the submanifolds in the data space.

**Lemma 2.** *For each submanifold $\mathbb{S}_i \in \mathbb{G}_{\backslash \rho}$, $\exists \mathbb{V}_j \in \mathbb{Z}_{\backslash \rho}$ such that $G(\boldsymbol{z}) \in \mathbb{S}_i \ \forall \ \boldsymbol{z} \in \mathbb{V}_j$. Likewise, for each submanifold $\mathbb{V}_j \in \mathbb{Z}_{\backslash \rho} \ \exists \ \mathbb{S} \in \mathbb{G}_{\backslash \rho}$ such that $G(\boldsymbol{z}) \in \mathbb{S}_i \ \forall \ \boldsymbol{z} \in \mathbb{V}_j$.*

**Theorem 2.** *Let $\rho$ be a scalar such that $q_G(G(\boldsymbol{z})) > \rho \ \forall G(\boldsymbol{z}) \in \mathbb{X}$ and $q_G(G(\boldsymbol{z})) \leq \rho \ \forall G(\boldsymbol{z}) \notin \mathbb{X}$. Then, for each submanifold $\mathbb{M}_i \in \mathbb{X} \ \exists \mathbb{V}_j \in \mathbb{Z}_{\backslash \rho}$ such that every point in $\mathbb{V}_j$ is mapped to a point in $\mathbb{M}_i$. Additionally, for every submanifold $\mathbb{V}_j \in \mathbb{Z}_{\backslash \rho}$, every point in $\mathbb{V}_j$ maps to the same submanifold $\mathbb{M}_i \in \mathbb{X}$.*

Theorem 2 follows directly from the preceding Theorem and Lemmas. In a nutshell, the above analysis shows that by removing points from the latent space that map to low-density points in the data space, *we can split the latent space of our pretrained generator $G$ into $k$ disjoint submanifolds (or clusters)*, where *each submanifold in the latent space corresponds to one of the submanifolds in the data space*. In other words, we now have achieved our important goal of introducing structure onto the prior latent space by removing data from certain regions of $\mathbb{Z}$.

The above findings refer to the concept of removing latent codes that are mapped to low-probability regions in the data space. Of course, this requires us to be able to identify which codes are mapped to low-probability regions. The following equation for the density of $G(\mathbf{z})$ allows us to do just that:

$$q_G(\boldsymbol{z}) = \frac{p_{\mathbf{z}}(\boldsymbol{z})}{\sqrt{det(\boldsymbol{J}_G^\top(\boldsymbol{z})\boldsymbol{J}_G(\boldsymbol{z}))}}, \tag{1}$$

where $\boldsymbol{J}_G(\boldsymbol{z}) = \frac{\partial G}{\partial \boldsymbol{z}}$ is the Jacobian of $G$ evaluated at $\boldsymbol{z}$. Equation 1 follows from the pushforward distribution between Riemannian manifolds and was is similarly utilized by recent works (Humayun et al., 2022a). Here, we utilize the expression in Equation 1 to identify points with a generated density less than $\rho$.

It is known that for high-dimensional data, a Gaussian distribution is well approximated by a uniform distribution over a sphere (Vershynin, 2018). Thus, for $\mathbf{z} \sim \mathcal{N}(\boldsymbol{\mu}, \Sigma)$, the generated density is approximately proportional to the following for the majority of realizations of $\mathbf{z}$:

$$q_G(\mathbf{z}) \propto \frac{1}{\sqrt{det(\boldsymbol{J}_G^\top(\mathbf{z})\boldsymbol{J}_G(\mathbf{z}))}}. \tag{2}$$

Equation 2 also holds for the case where $\mathbf{z}$ is from a uniform distribution.

Our above results imply a strategy for converting the original latent space $\mathbb{Z}$ of the pretrained generator $G$ into a structured latent space $\mathbb{Z}^*$. In short, first, we collect a large sample of $D$ points $\{z_i\}_{i=1:D}$ from $p_{\mathbf{z}}$. Second, we compute the density of each point according to Equation 1. Lastly, we remove each $z_i$ where $q_G(z_i) < \rho$, where $\rho$ is a hyperparameter that we chose.

A pseudocode algorithm for this process is given in Algorithm 1.

---

**Algorithm 1** Obtaining dataset from structured latent space $\mathbb{Z}^*$

---

**Require:** Pretrained generator $G$, sample size $D$, hyperparameter $\rho$
**Ensure:** Dataset of samples $\mathcal{Z}^*$ from structured latent space $\mathbb{Z}^*$
 1: Initialize empty set $\mathcal{Z}^*$
 2: Sample $\{z_i\}_{i=1:D}$ from $p_{\mathbf{z}}$
 3: **for** $i = 1$ to $D$ **do**
 4:     Compute density $q_G(z_i)$ using Equation 2
 5:     **if** $q_G(z_i) \geq \rho$ **then**
 6:         Add $z_i$ to $\mathcal{Z}^*$
 7: **return** $\mathcal{Z}^*$

---

### 3.2 Sampling From The Shattered Latent Space

We now need a strategy of performing online sampling from $\mathbb{Z}^*$ or from some specific submanifold $\mathbb{V}_i \subset \mathbb{Z}^*$, so as to perform controllable generation.

To that end, we first propose to cluster the new latent space $\mathbb{Z}^*$ so that we can identify the $k$ submanifolds it consists of. This can be achieved using one of the existing clustering algorithms. In our experiments we utilize k-means sampling due to its simplicity, and find it is sufficient to achieve our goal. Though more sophisticated approaches could equally be deployed, and they may yield even more robust clustering performance.

After clustering, the fitted clusterer $C$ maps each point $z \in \mathbb{Z}^*$ to a cluster label $\ell \in \{1, 2, \ldots, k\}$. Then, we propose to fit a *mapper* function $K : \{1, 2, \ldots, k\} \times \mathbb{J} \to \mathbb{Z}^*$ that allows us to sample from any of the k desired submanifolds in $\mathbb{Z}^*$. Specifically, the mapper $K$ takes in the index $i$ of the cluster we want to sample from along with a noise code $j$ and maps to a point $z \in \mathbb{Z}^*$ such that $z \in \mathbb{V}_i$.

As we do not want to modify the distribution of latent codes for each cluster, we parameterize $K$ to encourage $K_{i\,\#}p_{\mathbf{j}} = p_{\mathbb{Z}^*_{C=i}}$ for each cluster $i$. Specifically, we parameterize $K(i, \cdot)$ as a collection of Gaussian mixture models (GMM) fit to samples from $\mathbb{V}_i$. We utilizes GMMs as they are very computationally inexpensive. At the end, we can selectively sample from the $i$th sub-manifold by sampling from the GMM that was fit to the $i$th cluster in the latent space.

## 4 Related Work

**Disjoint manifold learning.** It is well known that typical generative models that are trained to map a connected $\mathbb{Z}$ to a disjoint data manifold are incapable of producing a disjoint distribution in their output space (Khayatkhoei et al., 2018; Salmona et al., 2022; Tanielian et al., 2020). Approaches such as Partition-Guided GANs (Armandpour et al., 2021), Disconnected Manifold Learning (Khayatkhoei et al., 2018), and MG-GAN (Dendorfer et al., 2021) address this issue by training multiple generators. Another standard approach is to utilize an unconnected latent space when training the generative model (Liu et al., 2022; Mukherjee et al., 2019; Gurumurthy et al., 2017). However, these approaches are applicable when training *new* generative models and do not address the problem of sampling from different modes given an *existing* generator. On the other hand, truncation approaches (Katzir et al., 2022; Tanielian et al., 2020; Azadi et al., 2018) *reject* certain sampled latent codes, which can allow a pretrained generative to sample from a disconnected latent space. Similarly, Latent Reweighting (Issenhuth et al., 2022) and Polarity Sampling (Humayun et al., 2022b) aim to reweight the latent distribution to sample latent codes that correspond to modes in the output space. While these approaches can be applied to pretrained generators, they do not typically allow for control over *which* mode is being sampled from at a given time.

**Converting unconditional models to conditional models.** Methods such as PromptGen (Wu et al., 2022) assume the availability of energy functions - such as pretrained classifiers - that can be utilized to learn a distribution of a certain class or attribute in the latent space. Latent Constraints likewise requires an auxilary classifier or user-provided reword function, and also requires training an additional GAN-like model on the latent space (Engel et al., 2018). This approach requires training additional invertable neural networks to sample from each of these learned distributions. HyperGAN (Laria et al., 2022) converts unconditional generative models into conditional models by training a Hyper Net on an auxiliary labeled training dataset containing instances of each class.

## 5 EXPERIMENTS

### 5.1 COMPARED METHODS

*Original Generator.* We fit $k$-means clustering on samples from the generator' original latent distribution $p_{\mathbf{z}}$. This acts as a baseline; the clusters found here should not meaningfully correspond to clusters or classes in the output space.

*Polarity Sampling.* This approach allows for sampling from modes or minima of the generated distribution through reweighting the original latent distribution (Humayun et al., 2022b).

*JBT Sampling.* This approach, proposed in (Tanielian et al., 2020), reweights the latent distribution according to the Frobenius norm of the generator's jacobian. Whereas our method reweights the latent distribution according to the *volume* change of the generator's transformation, JBT's approach corresponds to rescaling proportional to the length of the diagonal of the *diagonal* of the unit box after mapping through the generator. The diagonal provides less meaningful feedback on the change in density induced by the generator than our approach, which directly models the volume change.

For each method, we evaluate the quality of the clusters found through applying $k$-means on large samples from their induced latent distributions. We also fit a GMM on each cluster for each method and report metrics for sampling from clusters in this manner, reported as "*{method} + GMM*".

### 5.2 DATASETS

We compare all methods on three datasets: `MNIST` (Lecun et al., 1998), `FashionMNIST` (Xiao et al., 2017), and `Faces + Flowers`. The `Faces + Flowers` dataset is constructed by mixing together the `CelebA` (Liu et al., 2015) and `Flowers102` (Nilsback & Zisserman, 2008) datasets. We do this as the manifold of images of human faces and the manifold of images of flowers should be disjoint.

For each dataset, we pretrain a DCGAN (Radford et al., 2016) to match the data distribution. We use a latent dimension of 64 for each dataset. We likewise pretrain a convolutional classifier for each dataset to predict the classes in each. *The classifier is used only for evaluating the clusters found from each compared method.* We see Section A.2 in the appendix for more details on the architecture choices.

### 5.3 COMPARING CLUSTERS ON SAMPLES DIRECTLY FROM THE INDUCED LATENT SPACES

In this first experiment we analyze the quality of the clusters found when applying $k$-means clustering to the latent space induced by each method. For each method, we sample 20,000 latent codes from their induced latent space. We then cluster the latent codes using $k$-means clustering. Note that we do *not* fit a mapper function $K$ to sample from these clusters in this experiment; we instead perform our analysis directly on the latent codes and their cluster assignment.

For each cluster, we generate the corresponding output samples by passing the instances in each cluster through the pretrained generator. We then use the pretrained classifier to provide class labels for each instance in each cluster. Ideally, each cluster will be strongly correlated with one and only one class - under the assumption that each class lies on a distinct submanifold in the data space.

To measure the correlation between cluster assignments and class labels, we utilize four metrics: *Homogeneity*, *Completeness*, *V Measure* (Rosenberg & Hirschberg, 2007), and *Adjusted Mutual Information (MI)* (Vinh et al., 2010). *Homogeneity* is maximized when each cluster contains instances

Table 1: Comparative study of the cluster quality found from each method. Higher numbers are better for each metric. Best performance is bolded.

|  | Metric | Original Generator | Polarity Sampling | JBT Sampling | LASH (Ours) |
|---|---|---|---|---|---|
| MNIST | Homogeneity | $0.0374 \pm 0.0044$ | $0.1657 \pm 0.0069$ | $0.3786 \pm 0.0147$ | $\mathbf{0.4105 \pm 0.0118}$ |
|  | Completeness | $0.0375 \pm 0.0044$ | $0.1775 \pm 0.0077$ | $0.3793 \pm 0.0147$ | $\mathbf{0.4120 \pm 0.0119}$ |
|  | V Measure | $0.0374 \pm 0.0044$ | $0.1714 \pm 0.0073$ | $0.3789 \pm 0.0147$ | $\mathbf{0.4113 \pm 0.0118}$ |
|  | Adjusted MI | $0.0198 \pm 0.0045$ | $0.1555 \pm 0.0074$ | $0.3675 \pm 0.0150$ | $\mathbf{0.4004 \pm 0.0121}$ |
| Fashion MNIST | Homogeneity | $0.0336 \pm 0.0051$ | $0.1723 \pm 0.0128$ | $0.1965 \pm 0.0090$ | $\mathbf{0.2297 \pm 0.0091}$ |
|  | Completeness | $0.0337 \pm 0.0051$ | $0.1761 \pm 0.0127$ | $0.1974 \pm 0.0091$ | $\mathbf{0.2312 \pm 0.0092}$ |
|  | V Measure | $0.0336 \pm 0.0051$ | $0.1742 \pm 0.0128$ | $0.1970 \pm 0.0091$ | $\mathbf{0.2305 \pm 0.0091}$ |
|  | Adjusted MI | $0.0159 \pm 0.0052$ | $0.1588 \pm 0.0130$ | $0.1822 \pm 0.0092$ | $\mathbf{0.2163 \pm 0.0093}$ |
| Face +Flowers | Homogeneity | $0.0020 \pm 0.0016$ | $\mathbf{0.1290 \pm 0.0065}$ | $0.1027 \pm 0.0074$ | $0.1275 \pm 0.0121$ |
|  | Completeness | $0.0032 \pm 0.0025$ | $0.1804 \pm 0.0087$ | $0.1716 \pm 0.0113$ | $\mathbf{0.2211 \pm 0.0170}$ |
|  | V Measure | $0.0025 \pm 0.0019$ | $0.1504 \pm 0.0074$ | $0.1285 \pm 0.0089$ | $\mathbf{0.1617 \pm 0.0140}$ |
|  | Adjusted MI | $0.0020 \pm 0.0019$ | $0.1501 \pm 0.0074$ | $0.1281 \pm 0.0089$ | $\mathbf{0.1613 \pm 0.0140}$ |

Table 2: Comparative study of the sample quality from the mapper function $K$. Higher numbers are better for each metric. Best performance is bolded.

|  | Metric | Original Generator + GMM | Polarity Sampling + GMM | JBT Sampling + GMM | LASH (Ours) + GMM |
|---|---|---|---|---|---|
| MNIST | Homogeneity | $0.0364 \pm 0.0049$ | $0.1494 \pm 0.0058$ | $0.3507 \pm 0.0142$ | $\mathbf{0.3781 \pm 0.0145}$ |
|  | Completeness | $0.0366 \pm 0.0049$ | $0.1584 \pm 0.0061$ | $0.3519 \pm 0.0143$ | $\mathbf{0.3801 \pm 0.0146}$ |
|  | V Measure | $0.0365 \pm 0.0049$ | $0.1538 \pm 0.0059$ | $0.3513 \pm 0.0142$ | $\mathbf{0.3791 \pm 0.0145}$ |
|  | Adjusted MI | $0.0188 \pm 0.0050$ | $0.1377 \pm 0.0061$ | $0.3394 \pm 0.0145$ | $\mathbf{0.3677 \pm 0.0148}$ |
| Fashion MNIST | Homogeneity | $0.0340 \pm 0.0045$ | $0.1649 \pm 0.0089$ | $0.1800 \pm 0.0124$ | $\mathbf{0.2107 \pm 0.0082}$ |
|  | Completeness | $0.0341 \pm 0.0045$ | $0.1677 \pm 0.0095$ | $0.1810 \pm 0.0125$ | $\mathbf{0.2122 \pm 0.0083}$ |
|  | V Measure | $0.0341 \pm 0.0045$ | $0.1663 \pm 0.0092$ | $0.1805 \pm 0.0125$ | $\mathbf{0.2115 \pm 0.0083}$ |
|  | Adjusted MI | $0.0163 \pm 0.0046$ | $0.1509 \pm 0.0094$ | $0.1654 \pm 0.0127$ | $\mathbf{0.1970 \pm 0.0084}$ |
| Face +Flowers | Homogeneity | $0.0031 \pm 0.0014$ | $0.1196 \pm 0.0073$ | $0.0932 \pm 0.0051$ | $\mathbf{0.1268 \pm 0.0096}$ |
|  | Completeness | $0.0048 \pm 0.0021$ | $0.1684 \pm 0.0093$ | $0.1533 \pm 0.0091$ | $\mathbf{0.2145 \pm 0.0116}$ |
|  | V Measure | $0.0037 \pm 0.0016$ | $0.1398 \pm 0.0081$ | $0.1159 \pm 0.0064$ | $\mathbf{0.1593 \pm 0.0107}$ |
|  | Adjusted MI | $0.0033 \pm 0.0016$ | $0.1395 \pm 0.0082$ | $0.1155 \pm 0.0064$ | $\mathbf{0.1589 \pm 0.0107}$ |

from only a single class, while *Completeness* is maximized if all instances of a given class are given the same cluster assignment. *V Measure* is their harmonic mean, and is thus maximized when there is a perfect correspondence between cluster assignment and class label. *Adjusted MI* measures the mutual information between cluster assignments and class labels, normalized to account for the MI generally increasing even for random assignments when the number of clusters grows.

Results are shown in Table 1. First, note that each method is an order of magnitude higher than the baseline method. Importantly, our method routinely outperforms all other compared methods across all metrics. The only case where our method is not the best performer (and is instead second best) is for the *Homogeneity* score on the `Face + Flowers` dataset, for which Polarity Sampling slightly outperforms LASH. However, LASH significantly outperforms Polarity Sampling on the *Completeness* metric for this setting. This indicates that while each cluster found after applying Polarity Sampling contains mostly instances from single classes, not every class is strongly correlated with some cluster. A likely explanation for this is that polarity sampling over-samples the mode (the face manifold in this case), and thus both clusters it finds mostly contain face images while the flower manifold does not have a cluster it is as strongly associated with.

## 5.4 COMPARING QUALITY OF MIXTURE MODELS FIT ON THE INDUCED LATENT SPACES

In this experiment, we evaluate our proposed approach of using a Gaussian Mixture Model (GMM) to easily sample from the clusters found in our shattered latent space. To this end, we take the clusters identified in the preceding experiment and fit a GMM on each cluster. We utilize a two-component

GMM for each cluster. We then sample instances from each GMM, and compare the correlation between GMM samples and classes in the output space. If the clusters are correlated with classes or submanifolds in the output space (which is our goal), then the GMM samples should likewise be correlated with these classes if they -model the clusters well. To compare our approach with the state-of-the-art, we likewise fit GMMs on the latent space induced by each compared approach. We utilize the same metrics as in our previous experiment above.

Results are shown in Table 2. LASH again outperforms all other compared methods. We note that the performance of each method using the GMM to obtain samples is close to the corresponding performance in the last experiment, where the analysis was performed directly on the latent space without using any mapper network. This indicates that the GMMs are able to well-model the submanifolds in the latent space of each method, implying that using a GMM to cheaply sample from any desired latent cluster is a promising approach that has only minimal performance costs.

## 5.5 PARAMETER STUDY

We analyze the impact of our two important hyperparameters on LASH's performance: the choice of $k$ for $k$-means clustering, and the amount of mass we chose to remove (analogous to our choice of $\rho$). We utilize `MNIST` for this experiment. Figure 3a shows the average cluster entropy of LASH's clusters over a range of choices for number of clusters $k$. If each cluster is strongly associated with some class, the entropy should be low. We see that this is the case so long as the number of clusters is not very significantly over or under-estimated. Figure 3b shows both the average cluster entropy, as well as the average total entropy across clusters, as a function of the amount of mass removed. We see that the average cluster entropy decreases significantly as more mass is removed, indicating each cluster becomes more strongly correlated with a class. However, the average class entropy likewise decreases for the distribution of classes *across* clusters - indicating that as more mass is removed, we start to lose more instances of certain classes causing an increase in class imbalance. We see that removing around 20% of the mass results in minimal loss of overall diversity, while still having the desired property of a low average cluster entropy.

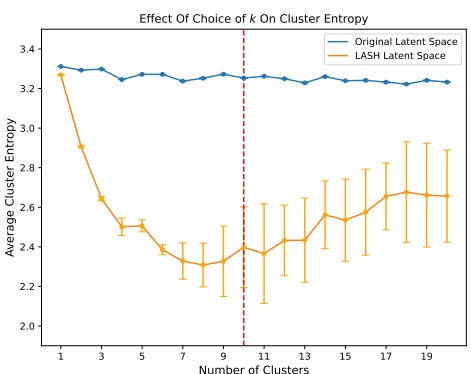
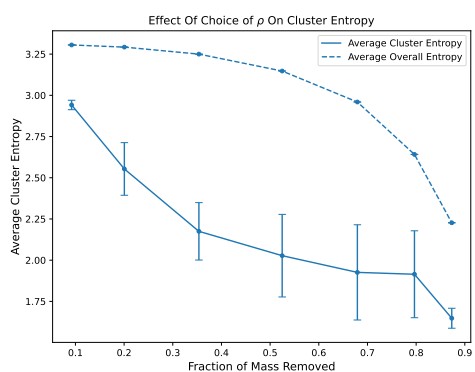

(a) Parameter study for number of clusters $k$

(b) Parameter study for amount of mass removed.

Figure 3: Parameter study for LASH.

## 6 CONCLUSION

In this work we have proposed *LASH*, a novel approach for converting pretrained unconditional generators into conditional generative models. Our approach is based on the idea that removing latent codes that are mapped to low-probability regions between submanifolds in the data space *shatters* the latent space into disjoint submanifolds, which we can then cluster and selectively sample from by utilizing helper mapper functions. Notably, we do not require retraining the generative model or any new deep network. LASH can be easily applied to any pushforward generative model that maps from a lower dimensional latent space with a known distribution into a high-dimensional data space. Our experimental analysis shows that the clusters found using LASH are highly correlated with classes in the output space.

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

# A APPENDIX

## A.1 HYPERPARAMETERS

*Polarity Sampling.* Polarity Sampling is determined using a parameter $\rho$ where as $\rho$ goes to $-\infty$ modes are sampled from increasingly often, and antimodes are sampled from more as $\rho \to \infty$, with the original generative distribution corresponding to $\rho = 0$. As we aim to sample from the on-manifold region of the output space, we set $\rho$ to a negative value such as to sample from modes rather than low-probability regions. Specifically, we set $\rho = -0.1$.

*Training Details.* For MNIST and Fashion MNIST, we trained each network for 50 epoch with a batch size of 64. For Celeba + Flowers we trained for 50 epochs with a batch size of 16. For all datasets and models, we used the Adam optimizer (Kingma & Ba, 2017) with learning rate set to 0.001 and betas=(0.5, 0.999). All models were implemented in PyTorch (Paszke et al., 2017).

## A.2 NETWORK STRUCTURES

For MNIST and FashionMNIST, we utilized the following network structures:

**Generator**

```
----------------------------------------------------------------
     Layer (type)               Output Shape
================================================================
  ConvTranspose2d-1           [-1, 256, 4, 4]
     BatchNorm2d-2            [-1, 256, 4, 4]
            ReLU-3            [-1, 256, 4, 4]
  ConvTranspose2d-4           [-1, 128, 8, 8]
     BatchNorm2d-5            [-1, 128, 8, 8]
            ReLU-6            [-1, 128, 8, 8]
  ConvTranspose2d-7           [-1, 64, 16, 16]
     BatchNorm2d-8            [-1, 64, 16, 16]
            ReLU-9            [-1, 64, 16, 16]
 ConvTranspose2d-10           [-1, 1, 32, 32]
           Tanh-11           [-1, 1, 32, 32]
================================================================
```

**Discriminator**

```
----------------------------------------------------------------
     Layer (type)               Output Shape
================================================================
          Conv2d-1            [-1, 64, 16, 16]
       LeakyReLU-2            [-1, 64, 16, 16]
          Conv2d-3            [-1, 128, 8, 8]
     BatchNorm2d-4            [-1, 128, 8, 8]
       LeakyReLU-5            [-1, 128, 8, 8]
          Conv2d-6            [-1, 256, 4, 4]
     BatchNorm2d-7            [-1, 256, 4, 4]
       LeakyReLU-8            [-1, 256, 4, 4]
          Conv2d-9            [-1, 1, 1, 1]
       Sigmoid-10            [-1, 1, 1, 1]
================================================================
```

**Classifier (For Evaluation)**

```
----------------------------------------------------------------
```

```
        Layer (type)              Output Shape
================================================================
          Conv2d-1              [-1, 16, 28, 28]
           ReLU-2               [-1, 16, 28, 28]
       MaxPool2d-3              [-1, 16, 14, 14]
          Conv2d-4              [-1, 32, 14, 14]
           ReLU-5               [-1, 32, 14, 14]
       MaxPool2d-6               [-1, 32, 7, 7]
         Linear-7                    [-1, 10]
================================================================
```

For CelebA + Flowers we utilized the following networks:

**Generator**

```
----------------------------------------------------------------
        Layer (type)              Output Shape
================================================================
  ConvTranspose2d-1             [-1, 512, 4, 4]
     BatchNorm2d-2              [-1, 512, 4, 4]
            ReLU-3              [-1, 512, 4, 4]
  ConvTranspose2d-4             [-1, 256, 8, 8]
     BatchNorm2d-5              [-1, 256, 8, 8]
            ReLU-6              [-1, 256, 8, 8]
  ConvTranspose2d-7            [-1, 128, 16, 16]
     BatchNorm2d-8             [-1, 128, 16, 16]
            ReLU-9             [-1, 128, 16, 16]
 ConvTranspose2d-10            [-1, 64, 32, 32]
    BatchNorm2d-11             [-1, 64, 32, 32]
           ReLU-12             [-1, 64, 32, 32]
 ConvTranspose2d-13             [-1, 3, 64, 64]
          Tanh-14              [-1, 3, 64, 64]
================================================================
```

**Discriminator**

```
----------------------------------------------------------------
        Layer (type)              Output Shape
================================================================
          Conv2d-1             [-1, 64, 32, 32]
       LeakyReLU-2             [-1, 64, 32, 32]
          Conv2d-3            [-1, 128, 16, 16]
       LeakyReLU-4            [-1, 128, 16, 16]
     BatchNorm2d-5            [-1, 128, 16, 16]
          Conv2d-6             [-1, 256, 8, 8]
       LeakyReLU-7             [-1, 256, 8, 8]
     BatchNorm2d-8             [-1, 256, 8, 8]
          Conv2d-9             [-1, 512, 4, 4]
      LeakyReLU-10             [-1, 512, 4, 4]
    BatchNorm2d-11             [-1, 512, 4, 4]
         Conv2d-12              [-1, 1, 1, 1]
        Sigmoid-13              [-1, 1, 1, 1]
================================================================
```

**Classifier (For Evaluation)**

```
----------------------------------------------------------------
```

```
     Layer (type)               Output Shape
================================================================
        Conv2d-1              [-1, 64, 32, 32]
     LeakyReLU-2              [-1, 64, 32, 32]
        Conv2d-3             [-1, 128, 16, 16]
     LeakyReLU-4             [-1, 128, 16, 16]
   BatchNorm2d-5             [-1, 128, 16, 16]
        Conv2d-6              [-1, 256, 8, 8]
     LeakyReLU-7              [-1, 256, 8, 8]
   BatchNorm2d-8              [-1, 256, 8, 8]
        Conv2d-9              [-1, 512, 4, 4]
    LeakyReLU-10              [-1, 512, 4, 4]
  BatchNorm2d-11              [-1, 512, 4, 4]
       Conv2d-12               [-1, 1, 1, 1]
      Sigmoid-13               [-1, 1, 1, 1]
================================================================
```

## A.3 PROOFS

### A.3.1 PROOF OF LEMMA 1

*Proof.* Assume that $G$ is not $k$-$\rho$-*disconnected*; then, $\mathbb{G}_{\backslash \rho}$ consists of $h$ distinct sub-manifolds, where $h < k$ or $h > k$. If $h < k$, then there was a sub-manifold in the data space with region for which $G$ did not place a density of at least $\rho$. However, this is not possible as it breaks the combination of Assumption 2 and Assumption 3. If $h > k$, then there is a subset outside of the support of $p_{\mathbf{x}}$ where $G$'s density is greater than $\rho$. This also breaks Assumption 2. Thus, Lemma 1 is proven by contradiction. $\square$

### A.3.2 PROOF OF THEOREM 1

*Proof.* By Lemma 1 we know that $\mathbb{G}_{\backslash \rho}$ consists of $k$ distinct sub-manifolds. Since by Assumption 1 $G$ is a continuous injective function, it's domain must also consist of $k$ distinct sub-manifolds. $\square$

### A.3.3 PROOF OF LEMMA 2

*Proof.* Due to $G$ being a continuous function, each $\mathbb{V}_i$ must be mapped to a single sub-manifold in the output space. Further, each of the $k$ output sub-manifolds requires there to be at least one sub-manifold in the latent space that is mapped to it. Since there are $k$ sub-manifolds in each space, there must be a one-to-one correspondence between them. $\square$

## A.4 DISTRIBUTION OF CLASSES IN LASH CLUSTERS

Results shown for MNIST:

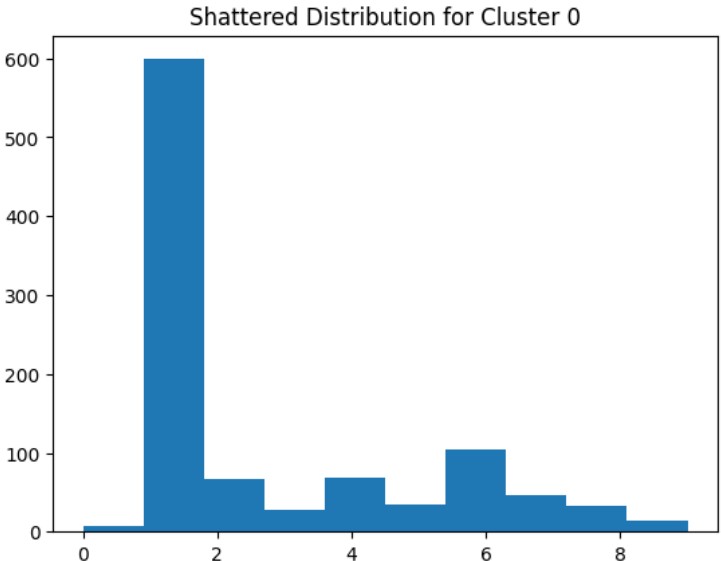

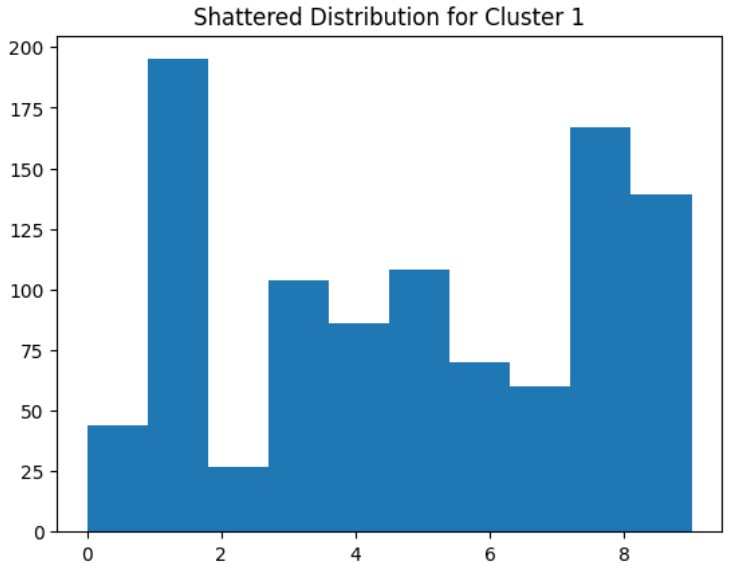

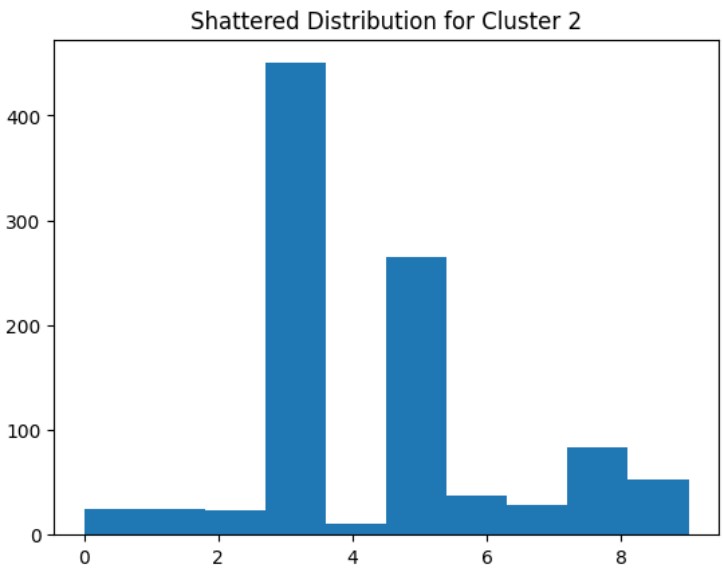

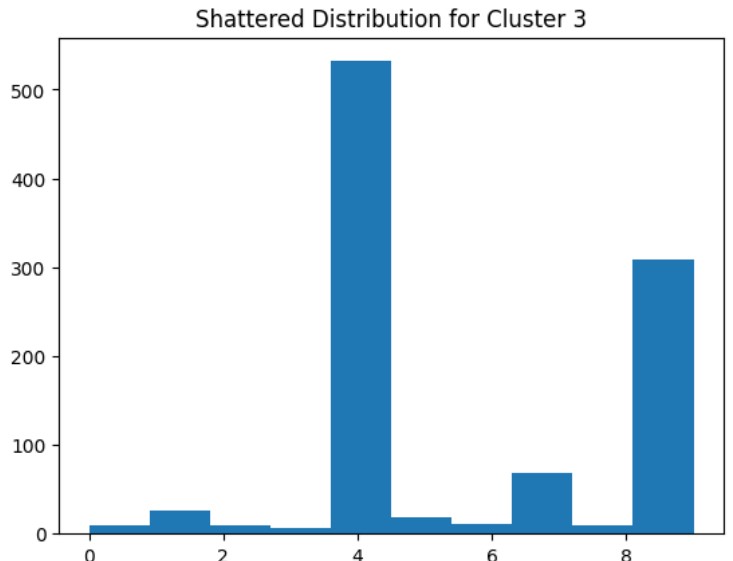

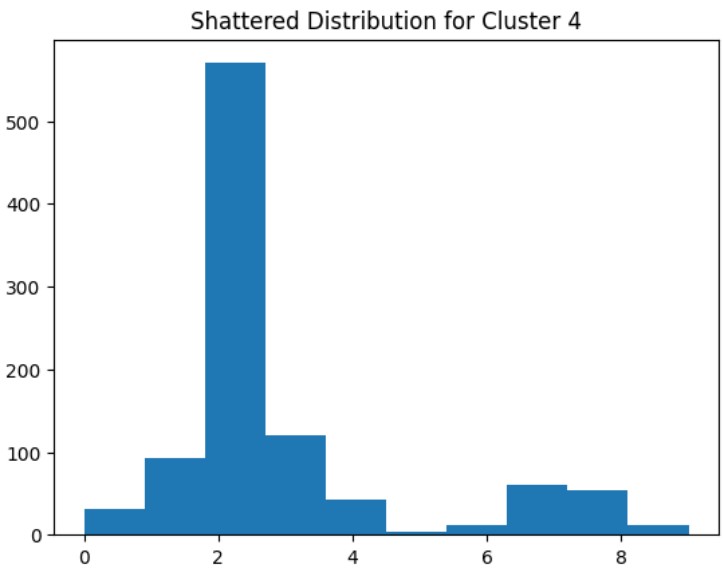

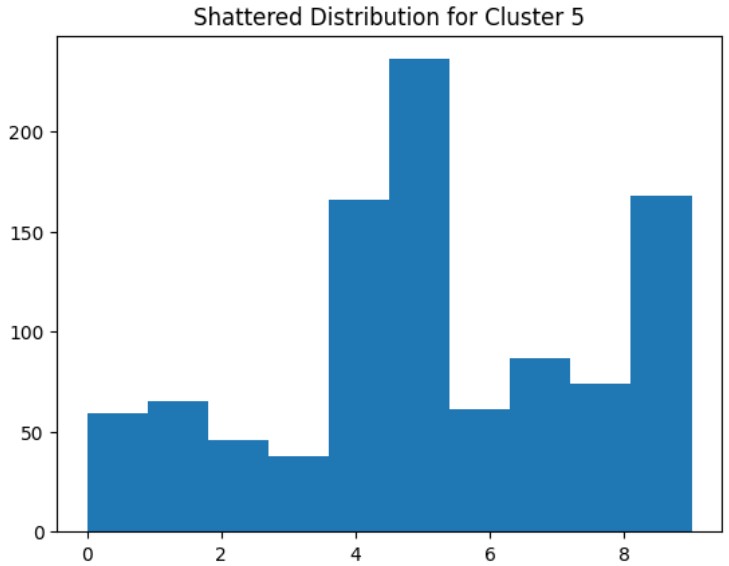

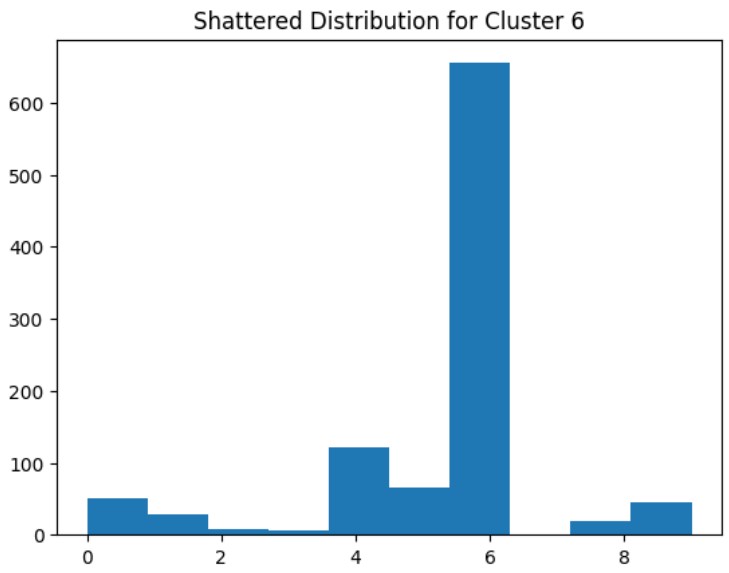

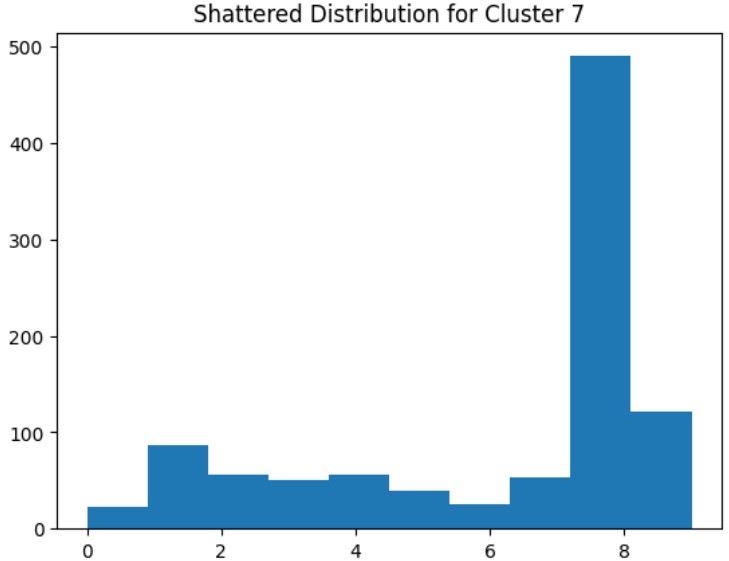

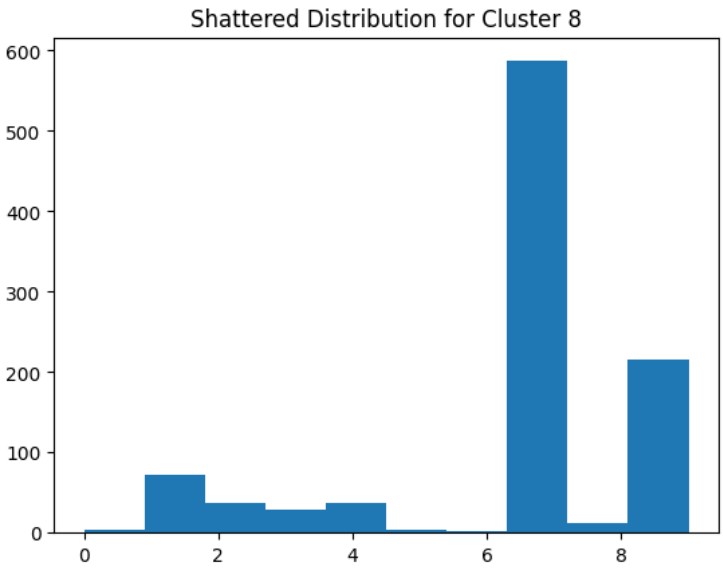

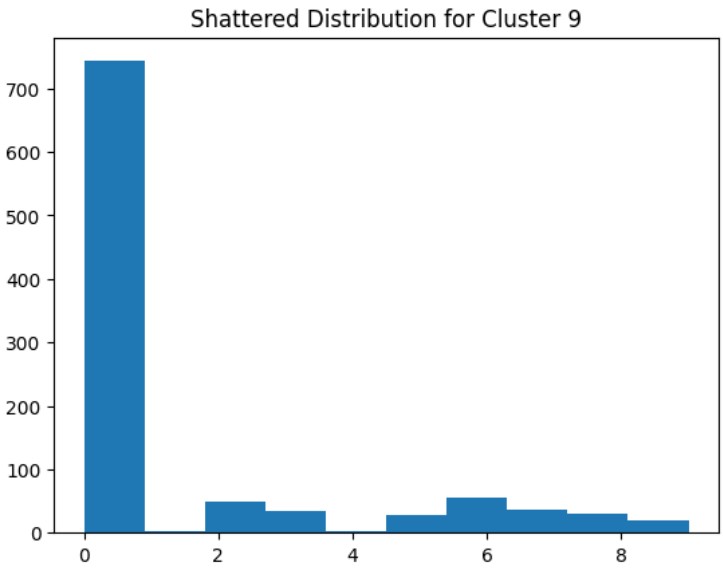

