# OpenReview forum: "Latent Shattering: Turning Unconditional Pretrained Generators Into Conditional Models By Imposing Latent Structure"
_ICLR.cc/2024/Conference — Submitted to ICLR 2024_

### Official Review · Reviewer_3NcT · 2023-10-30

**Soundness:** 1 poor
**Presentation:** 4 excellent
**Contribution:** 3 good
**Rating:** 5
**Confidence:** 3

**Summary:**

This paper's work arises from the idea that a good proxy for class labels in unconditional image generation is to identify the modes of the latent space of a pretrained generator. Given the assumption that the data generation probability concentrates on a set of sub-manifolds and the generator assigns a non-zero probability to space in between manifolds, the authors propose an unsupervised method to identify and learn to sample latent modes that correspond 1-to-1 to data classes.

**Strengths:**

- The paper demonstrates that the proposed method outperforms other latent clustering techniques, as evidenced by the results presented in Tables 1 and 2.
- The authors have succeeded in presenting their research in a clear and intuitive way.
- Related work connections are relevant, and the design choice is justified and sound.
- The paper's mathematical framework is transparent, and the authors have succeeded in presenting their proposed idea with an uncomplicated, yet thorough, exposition.

**Weaknesses:**

- The paper could benefit from a comparative analysis from methods mentioned in "Converting unconditional to conditional methods." of Related work, as it would demonstrate the proposed method's superiority beyond manifold learning.
- The absence of a comparison with any method on real-world images is a significant drawback. Real-world images more sophisticated than CelebA + Flowers102 dataset often introduce complexities not found in simpler or synthetic datasets, and such a comparison would help assess the method's practical applicability.
- The paper misses an opportunity to enhance its credibility by not incorporating the latest GAN architectures like StyleGAN. Analyzing clustering in the W space of StyleGAN could yield valuable insights and broaden the scope of the paper.
- Another limitation of the study is the absence of a dedicated image quality metric for evaluating the performance of image generation models. In particular, the inclusion of, e.g., inter-class or intra-class Frechet Inception Distance (FID) as a quantitative assessment would greatly enhance the comprehensiveness of the evaluation and comparison between different models.
- The omission of larger-scale datasets, like ImageNet or similar, is a missed opportunity to assess the method's scalability on more complex, mode-populated latent manifolds. An in-depth analysis on larger datasets would provide valuable insights into the method's performance under different conditions.
- While the parameter study is performed on small datasets, it is essential to expand this analysis to real-world datasets. A broader study would better reflect the method's adaptability to real-world scenarios and provide more comprehensive insights.

**Questions:**

- I noticed no use of single domain datasets, i.e., only faces. Would clustering on that allow fine-grained attribute control (e.g., hair style, smile, etc.) in that case? Analysis on that could enrich the paper's practical relevance.

The rest of my questions and suggestions are stated in the Weaknesses section. The method proposes a valuable idea, but further experimentation would be needed to ensure the practical relevance of the work, particularly the use of more complex datasets and comparison to other unconditional-to-conditional methods.

---

### Official Review · Reviewer_gFLe · 2023-10-30

**Soundness:** 3 good
**Presentation:** 3 good
**Contribution:** 2 fair
**Rating:** 3
**Confidence:** 4

**Summary:**

This paper proposes LASH, a method to turn a pre-trained unconditional GAN (or any pushforward generative model) into a "class"-conditional model in a fully unsupervised way, without access to a classifier. LASH first shatters the latent space: many samples from the prior are drawn, and those which correspond to low-probability images according to the generative model (as measured by the injective change-of-variable formula) are dropped. The remaining samples are then clustered through k-means, and a Gaussian mixture model (GMM) is fit to each cluster. To sample conditionally from a given cluster, LASH simply samples from the corresponding GMM and decodes using the pre-trained generator.

Overall, I think the problem tackled in this paper is interesting, and certainly original, albeit not of immediate practicality. I also think the authors present a simple and sensible approach to tackle the problem, which seems to perform well on the evaluated metrics. That being said, I believe the paper needs a much more thorough empirical evaluation. I will increase my score if the authors adequately address weaknesses 4 and 5 outlined below.

**Strengths:**

1. The problem being considered here is novel, and I think quite interesting.

2. The method proposed by the authors makes sense, and the fact that the authors consider simply using k-means on the latent space of the generator does provide evidence supporting that shattering the latent space helps.

3. The paper is easy to follow.

**Weaknesses:**

4. The paper aims to have a fairly formal mathematical presentation (which in it of itself is actually a good thing), but omits some relevant discussions and uses extremely strong assumptions -- and it hides the strength of the assumptions. I'll detail what I mean below, but overall this should either be addressed more carefully, the assumptions should be relaxed to strengthen the theory, or the paper should simply not attempt to be so formal:

(a) First, $p_G$ is not a density with respect to the Lebesgue measure. This should be clarified, as it implies that $p_G(x)$ can be non-zero only when $x \in \mathbb{G}$.

(b) I think assumption 2 is way too unrealistic. Because of (a), assumption 2 is implicitly assuming that the support of $p_x$ is completely contained in $\mathbb{G}$.

5. The experiments are hugely lacking:

(a) There is no qualitative comparison between methods. It would be valuable to actually show conditional samples from LASH and from the baselines.

(b) There is also no study of how LASH affects the overall quality of the generative model. I believe the authors show verify that LASH is not ruining the quality of generated samples. A way to do so would be as follows: sample unconditionally from LASH (by sampling a cluster with probability proportional to its size, sample from the corresponding GMM, and then apply the generator) and compute some metric of sample quality like FID. Then, compare this FID to that of the pre-trained GAN.

Finally, some minor points:

- Please update the bibliography file to cite published versions of papers whenever these exist. For example, Kingma & Welling's VAE paper is an ICLR 2014 paper, not an arxiv preprint; and ADAM is also an ICLR paper (these are the 2 examples that I noticed, but please go through all the citations).

- The first sentence in the "State-of-the-Art" paragraph is not properly conjugated ("existing work... have"; make "work" plural, or use "has").

- Missing period at the end of footnote 1.

- Figure 1 would be clearer if the shattered samples on latent space were actually a subset of the samples shown on the right.

- "Contributions" paragraph should be "Contributions." for consistency with the rest of the paper.

- Third paragraph of sec 2: should $q_z$ be $p_z$?

- Assumption 1 ends abruptly and is incomplete.

- I think the $\in$ symbol in assumption 3 should be $\subset$.

- Please put algorithm 1 at the top of the page.

- Please define $\mathbb{J}$ in sec 3.2.

- Section 5.1: "jacobian" -> "Jacobian", and "diagonal of the diagonal"

**Questions:**

6. It is not clear why eq 2 is needed in the first place. What is the issue with having to evaluate the Gaussian density? Why approximate it?

7. I think the study of LASH as $k$ is changed in sec 5.5 is interesting, but could also be expanded further. Intuitively, I'd imagine that when $k$ is small, the clusters correspond to very broad concepts on an image, whereas as it increases, they become increasingly specific. Have you tried anything to assess this hypothesis?

---

### Official Review · Reviewer_QipQ · 2023-10-31

**Soundness:** 2 fair
**Presentation:** 3 good
**Contribution:** 1 poor
**Rating:** 3
**Confidence:** 4

**Summary:**

In this work the authors propose to convert an unconditional latent-based generative model (GAN/VAE) into a conditional one without retraining. In doing so, the proposed method assumes that a sub-manifold in the latent space can presumably correspond to the desired condition, and finds the said sub-manifold by shattering the latent space into several sub-manifolds. The authors also provide theorems behind the proposed sampling technique and some experiments regarding the sample quality.

**Strengths:**

While there are several issues I have as deleted in weaknesses below, I think this paper has several strengths that should be highlighted and encouraged.

The use of clustering in laten space, as well as using entropy asas a criterion for selecting the optimal value of $k$ in clustering, which is a pretty neat approach. Furthermore, the theoretical foundation of the proposed method is articulated clearly and appears okay.

**Weaknesses:**

I have several concerns regarding this submission, primarily related to the positioning of the paper, the methodology, and the experiments conducted.


# Overstatemnt as Conditional Models

In the work, the authors frame the proposed method as turning unconitonal generators into *conditonal models*. However, this appears to be a significant over-statement. The proposed method can not align itself with any arbitrary user- or data-defined conditions. Instead, it identifies the sub-manifold already already in the latet space, which is an intrinsic structure therein.

Contrastingly, a large body of works, to name a few  [1,2,3] (cited by the authors) and [8,9,10] (missed by the authors), are demonstrated the capacity to handle arbitrarly definied condition, through the use of energe function from learned or pre-trained models. These methods are not restricted by the intrinsic structure of a specific latent space, and are thus much more powerful in applicaitons. By comparison, the scope of this work under review seems considerably narrower.

I acknowledge the authors’ clarification that their method in this paper does not compete directly with those that uses energe based methods/losses . However that should still leadto a position without the current over-statement.

# Limited Contribution

The focus on conditions inherent in the latent structure seems to restrict the potential contributions of this paper. This limitation is twofold:

- Firstly, as mentioned above, the proposed method does not accomodate arbitily defined conditions, limiting its application. Also, it limits itself to GAN/VAE istand of diffusion models where conditioning on energy function based loss is useful.

- Secondly, considering related works involving finding latent structure, for example [5,6,7] which the authors already cite. The contribution may be limited and assessing it could be a challenge. This challenge is exacerbated by experimental concerns I raise below.

# Insufficient Expriments

Several issues in the experimental section hinders a substantianl emeprical assessment of this work in its current form:

- The proposed metrics are highly taied to cluster assignments, which is more directly to the design of the proposed, clustering based method, rather than universally applicable metrics for evaluating latent structures. More conventional metrics like classification results (accuracy, precision-recall curves) and FID would be more appropriate here.

- The paper lacks visual examples illustrating the conditions identifiedby the method, which prevents qualitative evaluation.

- The paper lacks results from more established models, such as StyleGAN{,2,3}, BigGAN. Also, the omission of diffusion models hurts the relavece of this work, althoug that may be intentional.

**Questions:**

How is exactly $\mathbf{J}_{G(z)}$ in Eq (1) computed? Do you assume G is invertible and doing so is commutable? Would that suggest the use of energy function should be a trivial extension?

---

### Official Review · Reviewer_twZp · 2023-11-01

**Soundness:** 2 fair
**Presentation:** 2 fair
**Contribution:** 1 poor
**Rating:** 3
**Confidence:** 4

**Summary:**

This paper proposes a method called Latent Shattering (LASH) to split the latent space of pretrained generative models into meaningful disjoint manifolds, by identifying and removing instances mapped between modes in the output space. This can tune a pretrained unconditional generative model into a conditional generative model in a completely unsupervised manner without an extra classifier in previous works. A series of experiments are performed to validate the efficiency of LASH.

**Strengths:**

This paper is overall well-written and tries to perform conditional generation in an unsupervised way.

**Weaknesses:**

1.	It seems that this paper does not propose anything particularly new. In my opinion, it just samples some latent points with high probability density and perform clustering via k-means or GMM. Moreover, the density estimation method has been used in the previous work [1].
2.	Besides, the motivation of performing clustering in the latent space is not clear. Obviously. There is no actual meaning of clustering labels and therefore it is not suitable to utilize them for conditional generations.
3.	A similar idea was proposed in [2] a long time ago.
4.	The experiments cannot validate the contributions mentioned in Section 1 and do not present any generated image to show the actual performance of the algorithm.
5.	There are some typos in Section 4. For example, the “user-provided reword function” should be the “user-provided reward function.”


[1] Humayun A I, Balestriero R, Baraniuk R. Polarity sampling: Quality and diversity control of pre-trained generative networks via singular values[C]//Proceedings of the IEEE/CVF Conference on Computer Vision and Pattern Recognition. 2022: 10641-10650.
[2] Engel J, Hoffman M, Roberts A. Latent constraints: Learning to generate conditionally from unconditional generative models[J]. arXiv preprint arXiv:1711.05772, 2017.

**Questions:**

1.	What’s the meaning and motivation of performing clustering in the latent space?
2.	What’s the real technical contribution of this paper?

---

### Meta-Review · Area_Chair_3kiy · 2023-12-06

**Metareview:**

The paper develops a method that aims to enable generation with certain style/type from a generative model of VAE or GAN type trained on general (i.e., unconditional) data. The considered ability seems to attract potential interest, and the proposed method sounds intuitive and inspiring. The mathematical description is clear, though flaws exist (I agree with Reviewer gFLe's comment on this).

Nevertheless, the paper does not seem to have posed the method in the very right way, as it is not to "allow the user to specify the class of the output" in the usual sense, but only to allow generating a not externally specified style/type that is automatically learned in an unsupervised way. The authors are also expected to discuss related work more comprehensively. Experimental results are not yet convincing, due to e.g. lack of visual quality examples, relevant metrics and relevant baselines, and that the tasks are relatively toy compared to contemporary interest.

**Justification For Why Not Higher Score:**

While the reviewers found the idea interesting and presentation clear, the mentioned weaknesses are significant which downgrade the quality below ICLR standard. Authors did not respond to those weaknesses.

**Justification For Why Not Lower Score:**

N/A

---

### Decision · Program_Chairs · 2024-01-16

Reject